# Polygenic Risk Scores Contribute to Personalized Medicine of Parkinson’s Disease

**DOI:** 10.3390/jpm11101030

**Published:** 2021-10-15

**Authors:** Mohammad Dehestani, Hui Liu, Thomas Gasser

**Affiliations:** 1Department of Neurodegenerative Disease, Hertie Institute for Clinical Brain Research, University of Tuebingen, 72076 Tuebingen, Germany; smdehestani@gmail.com (M.D.); lzu_liuhui@163.com (H.L.); 2German Center for Neurodegenerative Disease (DZNE), 72076 Tuebingen, Germany

**Keywords:** Parkinson’s disease, polygenic risk scores, personalized medicine

## Abstract

Parkinson’s disease (PD) is the second most common neurodegenerative disorder characterized by the loss of dopaminergic neurons. The vast majority of PD patients develop the disease sporadically and it is assumed that the cause lies in polygenic and environmental components. The overall polygenic risk is the result of a large number of common low-risk variants discovered by large genome-wide association studies (GWAS). Polygenic risk scores (PRS), generated by compiling genome-wide significant variants, are a useful prognostic tool that quantifies the cumulative effect of genetic risk in a patient and in this way helps to identify high-risk patients. Although there are limitations to the construction and application of PRS, such as considerations of limited genetic underpinning of diseases explained by SNPs and generalizability of PRS to other populations, this personalized risk prediction could make a promising contribution to stratified medicine and tailored therapeutic interventions in the future.

## 1. Introduction

Parkinson’s disease (PD) is the second most common neurodegenerative disorder characterized by the abnormal aggregation of the protein a-synuclein in the form of Lewy bodies and Lewy neurites and the degeneration predominantly of dopaminergic neurons of the midbrain. PD presents with cardinal motor symptoms including resting tremor, muscular rigidity, and bradykinesia as well as various non-motor symptoms like cognitive impairment [1]. A positive family history of PD is found in approximately 15% of patients and in 5–10% of cases inheritance follows a classic Mendelian pattern [2]. However, the vast majority of PD patients are sporadic and in those, causation is thought to be polygenic with environmental components. Consistent with the common disease-common variant (CDCV) hypothesis, PD overall genetic risk can be considered to be a consequence of the synergistic effect of a large number of common low-risk variants [3]. Tremendous progress has been made over the past decade, particularly with the advent of large genome-wide association studies (GWAS) that have been improving our ability to understand and define disease risk in sporadic PD by increasingly identifying these low-risk variants [4].

PD is incurable and imposes an enormous medical and societal burden and its prevalence is expected to rise [5]. To date, extensive research has been conducted to explore the etiology, progression, and ultimately the treatment and prevention of the disease. The apparent PD heterogeneity necessitates the personalized medicine concept, which postulates that various genetic and pathophysiological contributions may underlie distinct subgroups. This, in turn, has encouraged the search for targeted treatments, for example for subgroups of patients who have particular genetic mutations [6]. Beyond Mendelian mutations, i.e., rare variants with strong effects, efforts to quantify the joint effect of dozens of common genetic variants, and to develop predictive tools measuring this cumulative genetic load within each individual, are hoped to facilitate population stratification and identification of high-risk individuals. This personalized risk prediction may hold promise for the future by the means of stratified medicine and tailored therapeutic interventions. However, each such claim will require extensive investigation to justify its practical application (GBD 2016 Neurology Collaborators, 2019) [5,7].

So-called Polygenic scores (PRS) have been constructed through the compilation of genome-wide significant variants emerging from successive and ever-larger GWAS with the intention to capture the cumulative effect of many low to intermediate risk variants in a patient population. The first disease studied with the PRS method was schizophrenia. The researchers who performed the first schizophrenia GWAS constructed scores of risk propensity and then called them polygenic scores [8]. PRS are hoped to be a prediction and risk stratification tool that holds promise for identifying individuals with a higher predisposition to complex diseases such as schizophrenia or PD as well as providing insights into the biological basis and predicting age-dependent clinical outcomes [9,10].

Here, we used search terms including “Parkinson’s disease,” “polygenic risk scores,” “PRS,” and “polygenic scores” in the PubMed advanced search engine to access all papers to review the current PRS approaches and their applications in PD.

## 2. PRS Calculation and Data Interpretation

Single nucleotide polymorphisms (SNPs) are the most common type of genetic variation in humans. Their frequencies in a population are identified through genome-wide association studies. A genome-wide association study (GWAS) is an approach used in genetic research to associate specific genetic variations with a particular trait, for example, a disease. This method involves scanning the genomes numerous unrelated individuals with and without the disease and looking for statistically significant differences in the frequency of SNPs, given a strictly adjusted P-value threshold, which can be used to predict the presence of a disease (www.genome.gov (accessed on 1 August 2015)). The vast majority of SNPs are located in non-coding regions of the genome and are therefore not changing the amino acid composition of gene products. Rather, they are thought to be involved in the regulation of gene expression. These SNPs represent so-called Expressed Quantitative Trait Loci (eQTL) contributing to disease risk [11]. Most human traits are influenced by a large number of SNPs, each with small effects, along with the environment, and this genetic profile and its subsequent interplay with the environment renders each individual unique [12].

PRS are usually calculated as the sum of common variants (SNPs) weighted by corresponding effect size estimates and certain P-values derived from GWAS summary statistics data.

To obtain a reliable PRS capable of predicting both disease risk and continuous clinical outcomes, variables such as linkage disequilibrium (LD) and P-value thresholds must be considered when using SNP weights, i.e., GWAS betas. The logic behind this is to take into account the overlap of SNP weights, as each individual GWAS beta has some degree of overlap with neighboring SNPs and not taking this into account would lead to an overestimation of the predictive utility of PRS. How these variables are accounted for determines our method for PRS construction. There are several tools including those using standard clumping plus thresholding (C+T) such as PRSice and PLINK, and fancier dedicated tools including LDPred, PRS-CS, JAMPred, and Lassosum that model the LD by taking advantage of computational shrinkage strategies based on LD reference data [8,13,14,15,16,17]. In C+T, which is the basic method used by most publications to date, SNPs are clumped and prioritized at the locus with the smallest GWAS P-value so that the retained SNPs are largely independent of each other and thus their effects can be aggregated under the assumption of additivity. In turn, more advanced methods take all SNPs while accounting for the LD between them by applying shrinkage techniques to their weights. These shrinkage strategies help mitigate the inherently noisy nature of the weights due to the LD redundancy of SNPs. Put differently, each beta corresponding to a SNP may share some information in terms of LD with nearby SNPs, leading to double counting of the weights and ultimately overestimating the results. Nevertheless, for a whole range of diseases there is a relatively subtle difference between the conventional C+T and the shrinkage methods in terms of predictive power and accuracy of PRS models [18].

Regardless of the computational method, PRS analyses can be characterized by the two main input datasets they require: (i) base data consisting of GWAS summary statistics and (ii) target data, consisting of genotypes (often imputed) and phenotypes of individuals which should be independent of the GWAS samples as any overlap between base and target data can give rise to overestimation in final results. A practical solution to this end is often used in consortial meta-analyses, for example, the generation of “leave-one-out“ meta-analysis GWAS results [19], whereby each contributing study is excluded from the meta-analysis in turn [20].

To ensure generalizability of the results, the PRS analyses were performed in an independent target sample, referred to as out-of-sample prediction. The computational outputs include different plots which need to be interpreted correctly and carefully. A typical PRS study tests for an association between a PRS and the phenotype (disease status or clinical outcome) in the target data. This association can be measured with goodness-of-fit metrics and the effect size estimate between specific strata. Goodness-of-fit or explained variance is represented by incremental R2. Additionally, this is usually reported for case/control outcomes as Nagelkerke’s R2 which is a statistically adjusted R2, and since the case/control ratio is not equal to the disease prevalence, it should be adjusted in this respect on the liability scale [20,21]. Other PRS results from the typical C+T method include strata plots showing how trait values vary with increasing PRS or measuring increased risk folding for disease in individuals with the highest PRS. Density and violin plots are also commonly used to visualize the discriminatory power of PRS between cases and controls. The predictive accuracy of the PRS models as a binary target predictor can be assessed using Area Under the Receiver Operating Characteristic curve (AUC) analysis. The AUC can be interpreted as the probability that a case ranks higher than the control, and by analogy, the higher the AUC, the better a PRS model can discriminate between cases and controls [22]. Finally, the predictive value of PRS models should be also validated in a validation cohort (must be independent and is usually a subset of prediction cohort) in a process known as out-of-sample validation [20,23]. A typical PRS study workflow is depicted in Figure 1. 

### 2.1. PRS in PD Status Prediction

The primary goal of a polygenic score as a prognostication tool is to classify individuals according to their disease risk and predict disease status by distinguishing cases from controls. Several studies conducted in recent years using different GWASs have established various general polygenic score models and reported their ability to adequately discriminate patients with PD from neurologically normal individuals. The accuracy range of these predictions, i.e., Area Under the Curve (AUC) analysis varies across studies (Table 1).

### 2.2. PRS and PD Clinical Outcomes

Some studies have examined the associations between PD PRS and clinical outcomes, such as age at onset (AAO) and motor and non-motor function. It was hypothesized that sporadic PD cases with earlier AAO might carry a higher cumulative burden of genetic risk factors with relatively low effect sizes. In fact, several studies have confirmed that higher PRS is significantly associated with earlier AAO tendency [24,25,26,27,28,29]. Genes that can be assigned to the mitochondrial function and maintenance pathways have been shown to contribute to PD risk [30]. A study conducted by Billingsley et al. [31] established a mitochondria-specific PRS calculating effects of all PD risk variants within genes implicated in mitochondrial function. This study found, to the contrary, that higher mitochondria-specific PRS was associated with later AAO. Additionally, the largest PD AAO GWAS [32] showed that not all PD risk loci influence AAO with significant differences between risk alleles for AAO. These all indicate that overall PD risk and PD AAO may be caused by partially overlapped biological processes.

Motor dysfunction is the cardinal symptom of PD. The PRS was also found to be associated with faster motor decline, measured by the time from diagnosis to Hoehn and Yahr Scale stage 3 and change in Unified Parkinson’s Disease Rating Scale part III (UPDRS III) score, after adjusting sex and AAO [9,33]. Levodopa is the most effective treatment for PD motor symptoms, but long-time dopamine replacement treatment may cause levodopa-induced dyskinesias (LID) [34]. Eusebi et al. [35] reported that LID development was significantly correlated with higher PRS (HR = 1.39, 95% CI = 1.08–1.78), indicating the association between the aggregate burden of known genetic risk variants of PD and LID development. However, a recent study conducted by Liu et al. [36] did not find the association between PRS and motor progression and explained that disease initiation and progression might be driven by different genetics.

Cognitive impairment is also common in PD and an important non-motor symptom associated with quality of life and caregiver burden [37]. Though it is established that PD patients with GBA mutations usually present more severe cognitive decline compared with non-carriers, Paul et al. [9] found that cognitive impairment in PD is also linked to the polygenic load of common risk variants, i.e., higher PRS. A prospective and general population-based study by Adams et al. [38] investigated the association between mild cognitive impairment (MCI) and subsequent conversion to dementia and PRS of genetic variants for PD. It was found that PD PRS was associated with non-amnestic MCI as well. Liu et al. performed a longitudinal genome-wide survival study in 3821 PD patients to identify genetic variants associated with progression from PD to PD dementia (PDD). It was found that a novel RIMS2 locus (HR = 4.77, P = 2.78 × 10^−11^) was associated with the prediction of PDD, while PRS was not associated with cognitive progression.

Interestingly, a longitudinal study by Kusters et al. [39] reported that hallucinations among PD patients are associated with AD PRS, especially driven by APOE, but not formally significant in the statistical analysis with PD PRS after adjusting for confounders. PRS was also not associated with impulse control disorders (ICDs) in PD patients [40,41]. These results may support that PD-associated symptoms like hallucinations and ICDs, and PD itself have different genetic backgrounds.

### 2.3. PRS and Penetrance of LRRK2 and GBA

Mutations in the *LRRK2* gene are the most common cause of monogenic PD and also strong risk factors for sporadic PD [42]. *G2019S* is the most frequent mutation and has incomplete penetrance. The risk of *LRRK2 G2019S* mutation carriers developing PD was 28% at age 59 years, 51% at 69 years, and 74% at 79 years [43]. Iwaki et al. [44] analyzed 833 heterozygous *G2019S* carriers (including 439 PD) to investigate if a cumulative genetic risk affects the penetrance of PD among *G2019S* carriers. They found PRS (OR 1.34, *p* = 0.005) was significantly associated with a higher penetrance of the *G2019S* mutation, especially among younger carriers. This result is in line with the latest study by Lai et al. [45], which included 1879 *LRRK2* mutation carriers (1810 *G2019S* carriers and 776 cases). It conducted the GWAS of penetrance of PD in *LRRK2* mutation carriers and also its correlation with PD PRS. They found a significant *CORO1C* locus signal and PRS was also found to be a significant predictor of penetrance of *LRRK2* variants.

Heterozygous *GBA* mutations are another common genetic risk factor for PD. The penetrance of *GBA* variants is 10–30% and age-related [46,47]. A study by Blauwendraat et al. [48] reported that common variants in *SNCA* and *CTSB*, known PD risk loci, are associated with the penetrance of *GBA* and *PRS* and could also modify the penetrance. Given ongoing clinical trials focusing on *GBA* and *LRRK2* PD patients, identification of factors influencing penetrance of them could be used to stratify carriers and for personalized prevention.

### 2.4. PRS and Biomarkers

The pathological hallmark of PD is the accumulation of Lewy bodies composed mostly of aggregated α-synuclein (α-Syn) [49]. A meta-analysis shows that cerebrospinal fluid (CSF) levels of total α-syn is slightly decreased in PD cases compared with healthy controls, but it is not sufficient as a diagnostic biomarker [50]. Moreover, Alzheimer’s disease CSF biomarkers including total tau (t-tau), phosphorylated tau (p-tau), and amyloid-beta 42 (Aβ42) are also significantly lower in PD individuals [51]. Several studies have examined the associations between PD PRS and these CSF biomarkers. Two studies conducted by Ibanez et al., [24,52], and one study by Li et al. [25] did not find an association between PRS and CSF α-syn, while Lee et al. [53] found that higher PD PRS was associated with lower CSF α-syn. Notably, Ibanez et al. [52] found that PD PRS was correlated with CSF Aβ42 and PD cases with higher PRS present lower CSF Aβ42 levels, indicating that PD development and accumulation of Aβ42 in the brain may share similar pathways.

Interestingly, a recent study [54] investigated the associations between PD PRS and blood levels of 370 lipid species and lipid-related molecules. It revealed eight specific lipid species (e.g., arachidonic acid) were associated with PD PRS, which implies the involvement these lipids in PD etiology. In addition, Tirozzi et al. [55] reported that platelet distribution width (PDW), a measurement of the variability in platelet size distribution in the blood, was also related to PD PRS (rg [SE] = 0.080 [0.034]; *p* = 0.019). However, the functional meaning of PDW and its potential utility as a biomarker for PD remains to be clarified.

PD progression is known to be associated with nigrostriatal dopaminergic degeneration and dopamine transporter (DAT) scans can quantify striatal dopaminergic activity [56]. Lee et al. [53] investigated the longitudinal association between PD PRS and striatal dopaminergic activity measured by 123I-N-3-fluoropropyl-2-β-carboxymethoxy-3β-(4-iodophenyl) nortropane (123I-FP-CIT) SPECT on 335 PD cases. The authors established two PRS: (1) PRS-1: PRS including 27 risk SNPs and (2) PRS-2: PRS using 23 risk SNPs with minor allele frequency > 0.05. PRS-1 was not associated with striatal dopaminergic activity and PRS-2 was associated with a slower decline of activity, but the other 4/27 rare variants were associated with faster deterioration of activity. This suggested PD risk SNPs with different allele frequencies have heterogeneous effects on striatal dopaminergic degeneration and more studies are needed to further investigate the association between PRS and dopaminergic activity.

### 2.5. PRS in the Identification of Biological Pathways

In spite of the great success of continuing GWAS at identifying risk variants, many of the underlying molecular pathways and cellular processes involved in PD remain elusive. Several studies have attempted to construct pathway-specific PRS (cumulative effect of pathway-specific genetic variation on PD risk) to shed light on the PD-related biological pathways. Previous findings suggest that dysfunction in the endosomal membrane-trafficking pathway (EMTP) could contribute to PD pathogenesis [55], and Bandres-Ciga et al. [57] assessed the role of the EMTP comprehensively in the risk for PD. The authors constructed an EMTP-specific PRS using risk variants within 252 EMTP-related genes in a cohort involving 18,869 cases and 22,452 controls. The EMTP-specific PRS showed a 1.25 time increase of PD risk per standard deviation of genetic risk, providing powerful genetic evidence that the EMTP plays a role in PD etiology.

Studies have shown the autosomal recessive PD genes (e.g., *PINK1*, *PRKN*, and *DJ-1*) are associated with mitochondrial quality control system and mitophagy, implicating mitochondria in the etiology of monogenic PD. Billingsley et al. [31] comprehensively studied the role of genes regulating mitochondrial function in sporadic PD using mitochondrial-specific PRS in the same cohort. The authors constructed two gene lists based on the evidence of indicating relevant protein products relating to mitochondrial function. The “primary” gene list including 196 genes has the strictest evidence implicated in mitochondrial disorders. The “secondary” gene list including 1487 genes is implicated in mitochondrial function and morphology. The “primary” mitochondrial-specific PRS showed a 1.12-times increase of PD risk per standard deviation of genetic risk, while “secondary” PRS showed a 1.28-times increase. This study further provided evidence of the involvement of mitochondrial dysfunction in PD etiology.

Notably, a recent study by Bandres-Ciga et al. [21] applied PRS to a total of 2199 publicly available gene sets representative of canonical pathways to define the cumulative effect of pathway-specific genetic variation on PD risk. The training dataset comprising 7218 PD cases and 9424 controls was used to construct the PRS, while the testing dataset comprising 5429 PD cases and 5814 controls was used for validation. Besides identifying previously reported EMTP and mitochondrial pathways, this study also nominated some novel molecular pathways (e.g., chromatin remodeling and epigenetic mechanisms) contributing to PD etiology.

Bandres-Ciga et al. [58] also integrated the PRS approach and single-cell RNA sequencing data from 24 brain cell types to investigate in which cell types risk variants are active. They observed PD risk is associated with increased cell expression specificity in dopaminergic neurons, serotonergic neurons, hypothalamic GABAergic neurons, and neural progenitors, indicating that these cell types are essential for understanding PD-relevant biological pathways. Andersen et al. [59] applied the PRS approach in the same training and testing dataset to investigate heritability enrichment partitioned by cell type, focusing on immune and brain cells. The cell-type-PRS was constructed based on risk variants within open chromatin regions of the specific brain and immune cell types, as defined by ATAC-seq peaks [59]. Compared with other brain cell types, the author found microglial-PRS showed the strongest association with PD risk in both training and testing datasets. This study highlighted the role of microglial in PD etiology.

### 2.6. PRS for the Establishment of Stratified PD Trials

Using PRS to stratify patients by identifying high- and low-risk subgroups may help to conduct stratified trials that use medications that are effective in some forms of PD proportionate to genetic risk. It turns out that such stratified designs can potentially increase the efficiency of a trial. The use of genetic, clinical, imaging, or other molecular biomarkers to recruit patients who are more likely to respond efficiently to intervention is key to trial success and a central concept in stratified trials. This was exemplified in the relevant success attributable to the recruitment strategy of the aducanumab trial in 2015 for Alzheimer’s disease by the means of shift away from this strategy, potentially linked to fewer positive results in drug development (https://www.alzforum.org/therapeutics/aduhelm, accessed on 1 August 2015) [60]. Additionally, regarding pathway PRS, as the predictive potential of pathway PRS can extend the evidence of the pathways involved in PD pathogenesis, they can be used to identify patients at risk because of the disturbance of a particular pathway, i.e., a particular endophenotype, e.g., PD patients with severe impairment in the mitochondrial pathway, and then try to boost mitochondrial function in these patients by medications e.g., coenzyme Q10 known as “mitochondrial enhancer” [61]. 

### 2.7. Translating PRS onto the Absolute Scale

There is mounting interest in the clinical application of PRS in terms of interpreting the results to the people seeking to know about their genetic risk, though there have been few efforts in this specific respect. Developing handy tools that convert PRS onto the absolute scale, i.e., the probability an individual will develop the outcome [62], is a milestone in terms of confident interpretation of their results in the clinic. The absolute risk conferred by a given relative risk is determined by the predictive utility of the PRS that is standardized to a Z-score and the population prevalence of the phenotype. For example, an individual’s polygenic Z-score for disease may be 1.96, indicating their polygenic score is higher than 97.5% of an ancestry matched population.

However, at the moment, PRS can only be converted to the absolute scale if a validation sample is available, which imposes a major limitation on its use. Pain and colleagues recently developed a method to convert polygenic scores to the absolute scale for binary and normally distributed phenotypes. This method requires only the predictive accuracy (AUC) or the adjusted variance explained by the PRS (R2) and provides a practical choice for educational and clinical purposes [63].

## 3. Future Directions and Limitations

Although advancements in PD polygenic scoring have improved our ability to identify high-risk individuals, there are still caveats about using PRS in the clinic to select people for potential clinical follow-up and therapeutic intervention. These concerns surrounding the clinical implementation of PRS can be due to technical drawbacks in its construction and also subsequent psycho-social implications of its use.

In terms of PRS calculation, in addition to the noisy nature of SNP weights and their LD redundancy, there is a concern that we incorporate genetic variants that may not correlate perfectly with the causal factor, leading to uncertainty in the interpretation of results. In addition, as PRS is preferably a surrogate for SNP-driven heritability, it does not fully represent the entire picture of genetic architecture in PD and overlooks other contributors such as rare and structural variants, gene-gene, and gene-environment interactions; therefore, it must be interpreted carefully. Additionally, the fact that the scores so far have largely been calculated from eurocentric data, i.e., European ancestry GWAS, gives rise to the biased behavior of PRS and reduces its applicability in populations other than the European population. As a hint shedding light on the future direction of PRS application in stratified medicine, conducting large genome-wide studies in African populations could rapidly improve the accuracy of PRS for all populations [64].

When it comes to the use of PRS in the clinical setting, another caveat that requires scrupulous attention is that a PRS, like any other diagnostic uncertainty, must be interpreted contextually for the patients. Although feedback of genetic risk of complex disease in at-risk patients does not always result in significant self-reported negative behaviors, and some potentially positive behavioral changes have been noted [65], knowing genetic risk for a disease may convey a sense of insecurity in some people. Hence, the way PRS results are explained to the patients is of importance as well. On the other hand, future efforts to reduce this uncertainty by focusing on transparency in informing people about their PRS seem necessary. Looking at the example of other diseases such as coronary heart disease, Torkamani and his team developed MyGeneRank, an app that can calculate a person’s PRS for coronary heart disease from their genetic data from 23andMe, health data collected on mobile devices, and a series of questionnaires. Their aim was to understand how people respond to receiving the score and monitor any changes in health-related behavior thereafter [66]. In summary, however, PRS significantly improves genetic risk assessment at the individual level for PD patients, as the growing evidence suggests, it should be evaluated in the context of realistic expectations of what PRS can and cannot deliver [67]. Further research is also needed to ascertain how PRS can be effectively mainstreamed into clinical practice and more data, e.g., biomarker data, are needed to allow implementation of PRS in this context.

## Figures and Tables

**Figure 1 jpm-11-01030-f001:**
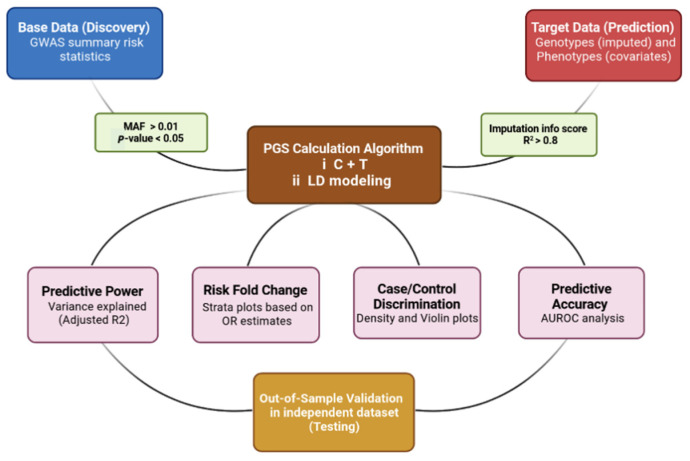
PRS calculation, results, and validation workflow. Summary risk statistics data should be filtered of rare variants (MAF > 0.01) and insignificant variants (*p*-value < 0.05). Additionally, target data-imputed genotypes should be filtered with a filter score (R^2^) of 0.8. In the PRS calculation box, C+T stands for clumping plus thresholding implemented in tools, e.g., PRSice software and LD modeling includes methods using shrinkage strategies implemented in tools, e.g., LDpred.

**Table 1 jpm-11-01030-t001:** Summary of general PRS studies that have performed PD risk prediction.

Study	Base Data	Number of SNPs	Target Population Size	PRS Calculation Tool	Predictive Accuracy (AUC)
Nalls (Nalls, Blauwendraat et al. 2019) [4]	2019 Nalls metaGWAS	90	37,688 cases, 18,618 UK Biobank proxy-cases (i.e., individuals who do not have Parkinson’s disease but have a first-degree relative that does), and 1.4 million controls	PRSice2	65%
Nalls (Nalls, Blauwendraat et al. 2019) [4]	2019 Nalls metaGWAS	1805	37,688 cases, 18,618 UK Biobank proxy-cases (i.e., individuals who do not have Parkinson’s disease but have a first-degree relative that does), and 1.4 million controls	PRSice2	69%
Ibanez (Ibanez, Dube et al. 2017) [24]	2014 Nalls metaGWAS	26	829 cases and 432 controls	Plink	Not published
Han (Han, Teeple et al. 2021) [10]	2019 Nalls metaGWAS	90	1654 PD Cases: 79,123 controls	LDpred	76%
Li (Li, Fan et al. 2019) [25]	2014 Nalls metaGWAS 2009 Satake GWAS 2017 Redensek metaGWAS	46	418 PD patients and 426 controls	Plink	61%
Foo (Foo, Chew et al. 2020) [26]	Asian GWAS	11	2536 PD cases and 21,840 controls	Plink	60.20%
Foo (Foo, Chew et al. 2020) [26]	Asian GWAS Nalls GWAS	11 + 90	2536 PD cases and 21,840 controls	Plink	63.10%

Abbreviations. SNP: single nucleotide polymorphism, PRS: polygenic risk scores, AUC: area under the receiver operating characteristic curve, GWAS: genome wide association studies, PD: Parkinson’s disease, LD: linkage disequilibrium.

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
