# Peer review of "Polygenic Risk Scores Contribute to Personalized Medicine of Parkinson’s Disease"

_jpm, 2021, doi:10.3390/jpm11101030_

Round 1

Reviewer 1 Report

Author summarized the current reports using the polygenic risk score approach in the PD field. It is well-organized and easy-to-follow manuscript which would help readers to update their knowledge. 

Author Response

Thank you so much for your comments. 

Reviewer 2 Report

This is a very interesting review (entitled "Polygenic Risk Scores Contribute to Personalized Medicine of Parkinson’s Disease"), summarizing studies on PRS and applications in PD.

A few comments:

  • In the Introduction part is referred that: “Here, we systematically review the current PRS approaches and their applications in PD.” If this is a systematic review then you should include in Methods the Search Strategy and Inclusion/Exclusion Criteria (and probably a flow chart of the literature search). A table with all included and excluded studies should also be included.
  • Figure 1. The usual p-value threshold for GWA SNPs is 5x10-8 and not 0.05.

Author Response

Thank you so much for your remarks. Regarding your first comment, we corrected that sentence and mentioned our search criteria that were used in the manuscript and since this was just a simple search based on certain terms we have not had any specific inclusion/exclusion criteria. Also regarding your second comment, I would like to clarify that this P=0.05 threshold is applicable to all SNPs from GWAS summary statistics to select the significantly associated SNPs with PD out of all discovered SNPs. This is applied just in case we aim to increase the power by relaxing the 5x10-8 cutoff. 

Reviewer 3 Report

This manuscript by Dehestani and colleagues is a concise, well-written, and slightly technical review on the possible uses of the very early science of polygenic risk scores in the field of Parkinson’s disease. At the moment, this approach has produced interesting but sometimes inconclusive and contradictory results; however, it looks promising for the future. Fairly, in the last paragraph, the authors highlighted the unresolved issues of this field, including the following: 1) The SNPs identified by GWAS are often not risk variants themselves but only tagging SNPs of a nearby “real” risk variant, 2) GWAS do not typically include genetic variations other than SNPs that may contribute to disease risk (e.g., CNVs), 3) the problem of low representation of ethnic minorities in GWAS, because tagging SNPs in a population can be different from those of a different population, 4) the possible unintended consequences of Parkinson’s disease PRS in the general population. I suggest to include some of these limitations in the abstract of the review. I have no major remarks to make.

Author Response

Thank you for your comments. I have already added some of the limitation remarks to the abstract.